# Satisfaction and continuance intention of blended learning from perspective of junior high school students in the directly-entering-socialism ethnic communities of China

Chunyu Li[1,2]*, Thanawan Phongsatha[1]

1 Graduate School of Business and Advanced Technology Management, Assumption University, Bangkok, Thailand, 2 International Cooperation and Exchange Center, West Yunnan University, Lincang, Yunnan Province, China

* 1402434006@qq.com

**Data Availability Statement:** All relevant data are within the paper.

**Funding:** The author(s) received no specific funding for this work.

## Abstract

### Background

Blended learning in DESEC only started after the introduction of the internet in recent 5 years. However, there is still no research paying attention to this region, because the area is remote and research subjects are not easily accessible. This article has potential application value in helping the government and educational institutions to make decisions on blended learning strategies supporting poverty alleviation through education in poor and remote areas and ethnic region. The study will be the first to examine satisfaction and continuance intention of blended learning in the DESEC.

### Objective

To identify junior high students' perception of satisfaction and continuance intention for blended learning in DESEC.

To identify the strongest factors affecting junior high students' satisfaction and continuance intention of blended learning in DESEC.

### Methods

A subsample of 635 junior high students participated online survey with consent of their parents verbally in computer room in schools under teacher's instruction. Data was coded and analyzed to generate descriptive statistics and inferential statistics. Structural equation model was used to evaluate the model of satisfaction and continuance intention of blended learning.

### Results

The level for evaluating students' agreement on each of item were interpreted "agree" (3.76–3.89). The model explained variances ($R^2$) of Continuance Intention, Satisfaction and Perceived usefulness were 0.665,0.766,0.718 respectively. Information quality, self-efficacy

**Competing interests:** The authors have declared that no competing interests exist.

and confirmation directly and indirectly contribute to junior high students' satisfaction with blended learning, which further confirmed their continuance intention of blended learning.

## Conclusion

Information quality was the strongest factor affecting the junior high students' continuance intention of using blended learning, while confirmation was the strongest factor affecting the junior high students' satisfaction of using blended learning in DESEC. Junior high students do not have a strong and distinct perception on satisfaction and continuance intention for blended learning in DESEC.

## Introduction

The Directly-Entering-Socialism Ethnic Groups (DESEG), also called Ethnic Groups with Cross-Stage Development, refers to the ethnic groups who were in the end of primitive society when the People's Republic of China was found [1]. The Communist Party of China implemented a special direct transition policy, which helped DESEG achieve a historic leap directly entering socialist society from primitive society [2]. Most of the directly-entering-socialism ethnic communities are located in barren mountainous areas, with poor natural conditions, relatively backward infrastructure construction such as transportation, sanitation, communication, and housing [3]. People's living standards are very low, and resource are relatively scarce [4]. Unity and progress of the ethnic minorities in this region is related to the harmony and stability of the China. It is a place that the Chinese government attaches great importance to.

Using modern information technology to drive the overall development of education in ethnic minority areas has become a useful experience in China's poverty alleviation work. At present, blended learning, a successful outcome of the combination of modern information technology and education, has been widely used in higher education, primary and secondary education, and has become a research hotspot in the field of educational technology in China. However, there is a huge gap between rural schools and urban schools in terms of the application technology, and a new imbalance in education is forming [5], letting alone schools in DESEC where technology application confronts challenges and hardships. Intergenerational updates of information technology equipment lag far behind urban schools [6]. Each administrative unit in different regions has their own independent data centers and application platforms, but resource cannot be interconnected and shared due to inconsistent construction standards, resulting in information isolation [7]. The lack of digital resource is a fatal shortcoming affecting the operation of online teaching in rural schools [8]. In addition, highly educated teachers or excellent teachers cannot be retained [9].

Blended learning in DESE only started after the introduction of the internet covering the schools in recent 5 years. But, the researches on blended learning dissertations are mainly concentrated in the field of higher education. There is almost no research pay attention to blended learning in DESEC of China due to the remoteness and not easily accessible population. Considering the equality and equity of education and the strategic position of this region, researchers want to break down such barriers. It sees that in recent years, the Chinese government has done a lot of work in training teachers' information literacy, which has achieved certain achievements. Equipment renewal requires a lot of money and investment that needs to be solved periodically. Therefore, this research focused on the student subjective sensation in

blended learning system. The study aimed at examining junior high students' satisfaction and continuance intention for blended learning in DESEC, so as to guide government and educational institutions in future decision-making on blended learning strategies.

## Materials and methods

### Previous research and research model

Self-Determination Theory (SDT) is a psychologically organismic and empirical theory that focuses on human behavior and individual development [10]. The theory explained that behavior continuum came from the intrinsic motivation, which means individual has intention to act for its inherent satisfactions [11]. To pursue better measures for explaining and predicting user acceptability of computers, Davis developed and validated the Theory of Acceptance Model (TAM), with Perceived Usefulness (PU) and Perceived Ease of Use (PEOU) [12]. The two variables are hypothesized as two primary relevance for user acceptance of information technology. Perceived Usefulness (PU) refers to the degree to which a person believes that using a particular system would enhance his/her job performance, and it is one of extrinsic motivators [13–15]. Combining Davis et al.'s formulation of TAM and Expectation-Confirmation Theory (ECT) [16,17] from the consumer behavior literature to illustrate I/S continuance, Bhattacherjee developed ECM. The model suggests that user' continence intention of IS depends on users' confirmation, PU as well as satisfaction [18]. Continuance intention refers to a person's psychological intention to do something consistently. People will act their previous behavior again with an approachable reinforcement [19]. Satisfaction is used to describe users' satisfactory feelings towards his or her adoption of a system [20]. Confirmation is the degree of users' perception of the congruence between expectation use and actual performance [18]. Perceived usefulness and satisfaction are powerful predictors of individuals' intention to use continence in B2C services. ECM is one of the most extensively utilized models on users' continuance intention of the information system. Hence, it leads to the hypothesis:

$H_{01}$: Perceived usefulness does not affect continuance intention of blended learning.

$H_{a1}$: Perceived usefulness affects continuance intention of blended learning.

$H_{02}$: Perceived usefulness does not affect satisfaction intention of blended learning.

$H_{a2}$: Perceived usefulness affects satisfaction intention of blended learning.

$H_{03}$: Satisfaction does not affect continuance intention of blended learning.

$H_{a3}$: Satisfaction affects continuance intention of blended learning.

$H_{04}$: Confirmation does not affect satisfaction of blended learning.

$H_{a4}$: Confirmation affects satisfaction of blended learning.

$H_{05}$: Confirmation does not affect PU of blended learning.

$H_{a5}$: Confirmation affects PU of blended learning.

Researchers expanded ECM with two external variables which modified PU to predict junior high students' behavior in blended settings [12]. Bandura proposed Social Cognitive Theory (SCT). He proved the predictive generality and explanatory of self-efficacy theory on illustrating human affect, motivation, and action [21]. Self-efficacy (SE) is level of individual confidence in one's ability to organize action to achieve specific tasks [22], and it affects what behaviors individual decide to perform [23]. Hence, it leads to the hypothesis:

$H_{06}$: Self-efficacy does not affect perceived usefulness of blended learning.

$H_{a6}$: Self-efficacy affects perceived usefulness of blended learning.

DeLone and McLean organized diverse research, introduced the quality constructs (information quality, system quality and service quality) and proposed an I/S success model [24]. Many previous research found information quality is the most critical factor, comparing with

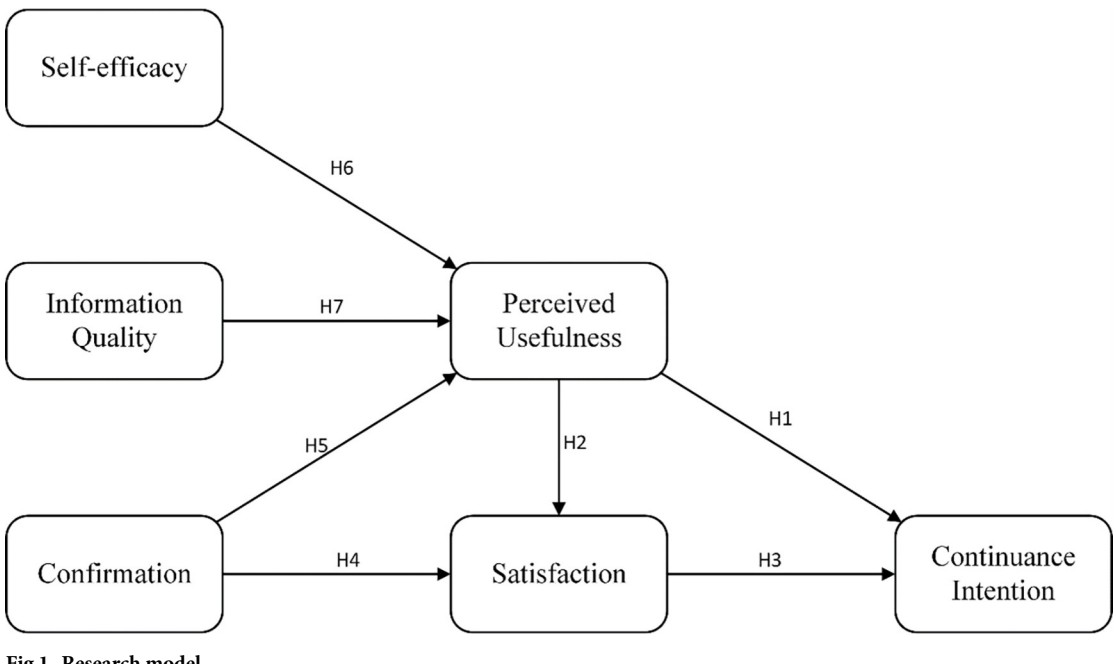

**Fig 1. Research model.**

system quality and service quality, to understand users' behavioral intention towards IS [13,25,26]. Information quality refers to the qualitative and desired characteristics of contents and form that the Information System (I/S) generates [24]. Hence, it leads to the hypothesis:

$H_{07}$: Information quality does not affect perceived usefulness of blended learning.

$H_{a7}$: Information quality affects perceived usefulness of blended learning.

Researchers sorted out satisfaction, self-efficacy, information quality, perceived usefulness, and confirmation based on the theories discussed above and literature review to describe the junior high school students' satisfaction and continuance intention of blended learning. The research model tried to facilitate analysis from psychological (satisfaction, self-efficacy, confirmation), technological (perceived usefulness, information quality), and behavioral factors (continuance intention) [27]. The research model is depicted in Fig 1.

## Scope of the research, design and target population

The characteristics of the population in this study was identified as students who were middle school students, were experiencing the blended learning, and living in the Direct-Entering-Socialism-Ethic Communities (DESEC). In order to conduct the study, the Cangyuan Va Autonomous County (short for Cangyuan, Yunnan province, southwest China) was selected. Because Cangyuan is typical representative of the DESEC and the key supported demonstration base for blended learning. The scope of the study was limited to recruit 3 schools in Cangyuan where students are able to access to online resources and take part in the classroom activities under the guidance of teachers, which required that schools were capable to provide internet and terminal equipment for blended learning. The students in K7 were too young to understand the questions well according to the researchers' previous data collection because the common method bias was serious. Hence, the target population of this study was high junior students in K8-K9 who had been experiencing blended learning at three middle schools in Cangyuan.

## Sample size and sampling technique

The sample size in this study was determined by using the Structural Equation Modeling. The anticipated effect size in this study was 0.2, and the desired level of statistical power was 0.8. According to 6 latent variables and 25 observed items, the recommended minimum sample size was 403. This study applied the judgement sampling and quota sampling for selecting the samples from the three middle schools in Cangyuan. In the end, a total of 635 valid questionnaires were received.

## Research instruments

The questionnaire of this study consisted of three main parts. The first part was screening questions, which helped researcher to filter the target population. The second part was items for measuring independent variables and dependent variable. For this part, the five-point Likert scale was used to measure these variables. The five-point Likert scale ranging from Strongly Disagree = 1; Disagree = 2; Undecided = 3; Agree = 4; and Strongly Agree = 5 was applied to examine factors related to students' continuance intention of blended learning. The third part is demographics demographic information (ethnicity, gender, blended learning experience, online study time, and occupation of guardians) of the respondents.

In addition, the Index of Item-Objective Congruence (IOC) [28] was conducted by inviting 3 qualified experts, who evaluated the items of variables to ensure the validity of the scale [29]. The study conducted a cut-off score to evaluate items in each of the construct: core 1 (I am sure the item can measure the construct), 0 (I am not sure the item can measure the construct), or -1 (I am sure the item cannot measure the construct). The items should be removed if IOC score lower than 0.5 level rating [30]. At last, 25 items (4 items mearing perceived usefulness [12,31,32]; 4 items mearing Satisfaction [31,33,34]; 4 items mearing Confirmation [35,36]; 4 items mearing Self-efficacy [36–38]; 5 items mearing Information Quality [31,39,40]; and 4 items mearing Continuance Intention [36,41,42] were validated shown as Table 1. In need of special note that the attention check question (IQ1*) of which meaning was opposite to the IQ1, i.e., "B-learning doesn't provide me sufficient contents I want" was designed to filter invalid respondents who were randomly choosing the answer without reading contents [43].

## Data collection

Participants were informed about the study content by their instructor and gave verbal assent in the classroom. All respondents were anonymous. The data collection procedure was organized in the computer room in the school. In this study, data collection was conducted through two stages, which were pilot test and main survey from September 2021 to January 2022. Researchers conducted pilot test to ensure the internal consistency reliability of the scale before the data collection process [44]. The Cronbach's alpha of each construct calculated by SPSS. Qualitative descriptors of satisfaction (α = .939), confirmation (α = .912), perceived usefulness (α = .866), self-efficacy (α = .854), information quality (α = .895), and continuance intention (α = .884) are satisfactory [21].

Guidance was designed for students when answering the questions. Respondents' answers were recorded and collected by WJX.cn online. Guide teacher were invited to make sure students correctly understand the meaning of each item. Thus, the researcher firstly trained relative teachers who was responsible to the survey. The questionnaire for students were distributed when the students were having their information technology course. Teachers emphasized that the students should finish the questionnaire by themselves without any communication among them. Students were asked to listen carefully to the teacher when answering the item under the teachers' administration.

**Table 1. Operationalization table.**

| Variable | Definition | Indicator | Source |
|---|---|---|---|
| **Perceived Usefulness** | The degree to which a student believes that blended learning would enhance his\her performance in study (Lee,2006) [15]. . | PU1: Blended learning enhances my learning effectiveness.<br>PU2: Blended learning can improve my learning performance.<br>PU3: Blended learning gives me greater control over my learning.<br>PU4: I find blended learning useful in my learning. | Almaiah et al. (2019) [31]<br>Cheng (2020) [14]<br>Alkhawaja et al. (2022) [32] |
| **Satisfaction** | The emotional perspective of the degree to which students' requirements have been fully satisfied by the blended learning system (Almaiah,2019 [31]). | SAT1: I am content with the performance of blended learning.<br>SAT2: I found choosing blended learning was a wise decision.<br>SAT3: I am happy with the functions provided by blended learning.<br>SAT4: I am satisfied with the overall experience of blended learning. | Mohamed et al. (2014) [33]<br>Almaiah et al. (2019) [31]<br>Warren et al. (2020) [34] |
| **Confirmation** | The degree of students' perception of the congruence between expectation and actual performance of blended learning (Bhattacherjee, 2001 [18]). | CON1: My experience with blended learning was better than what I expected.<br>CON2: The service level provided by blended learning was better than what I expected.<br>CON3: The information quality provided by blended learning was better than what I expected.<br>CON4: Overall, most of my expectations from blended learning were confirmed. | Joo et al. (2016) [35]<br>Yang et al. (2022) [36] |
| **Self-efficacy** | The level of individual confidence in one's ability to organize action to achieve specific tasks (Bandura, 1997 [22]). | SE1: I believe that I can use different blended learning software and systems to receive education.<br>SE2: I have the necessary knowledge to use software and systems provided by blended learning even if nobody is around to show me how to use it.<br>SE3: I am confident of using system provided by blended learning even if I have never used it before.<br>SE4: I am confident that I can overcome any obstacles when using the blended learning system. | Mohammadi (2014)<br>Vermunt& Donche (2017)<br>Yang et al. (2022) [36] |
| **Information Quality** | The qualitative and desired characteristics of contents and form that the Information System generates (Cheng, 2014) [13]. | IQ1: Blended learning provides me sufficient contents I want.<br>IQ1*: B-learning doesn't provide me sufficient contents I want.<br>IQ2: The difficulty level of the learning contents provided by the blended learning is appropriate.<br>IQ3: Blended learning provides me information is well formatted.<br>IQ4: Blended learning provides me timely information.<br>IQ5: Blended learning provides me new, updated learning contents. | Leon (2018) [39]<br>Almaiah et al. (2019) [31]<br>Alfaki et al. (2021) [40] |
| **Continuance Intention** | People will act their previous behavior again with an approachable reinforcement (Yan et al. 2021 [27]). | CI1: I intend to continue blended learning in the future.<br>CI2: I will learn more about using blended learning systems in the future.<br>CI3 I will frequently use system provided by blended learning on a regular basis in the future.<br>CI4 My intentions are to continue blended learning than any alternative means (traditional learning). | Ghazali et al. (2019)<br>Li et al. (2021)<br>Yang et al. (2022) [36] |

## Data analysis

After data collection, Statistical Package was used for statistical treatment to analyze and interpret data into meaningful results. IBM Statistical Package for the Social Science (SPSS) and AMOS version 23 software was applied in this study. Statistical treatment in this study included descriptive statistics and inferential statistics. The Cronbach's Alpha coefficient and the Corrected Item-Total Correlation (CITC) was analyzed to ensure the internal consistency of the questionnaire. The study applied confirmatory factor analysis (CFA) to measure the

validity. Before CFA, the study first performed exploratory factor analysis for checking correlation matrix, anti-image correlation matrix and the Kaiser-Meyer-Olikn (KMO) and the Bartlett's sphere by SPSS to make sure data were suitable for CFA. The study chose Maximum Likelihood (ML) for structural equation models (SEM) to evaluate the model. Statistical significance of this study was set at p<0.05.

## Results

### Demographic characteristics of respondents

Demographic characteristics of junior high students were shown in Table 2 (N = 635). A total of 635 valid questionnaires were obtained in this study, of which 341 students (53.7%) were from K8, and 294 students (46.3%) were from K9. There were 319 male students (50.2%), and 316 female students (49.8%). The sample size was composed of Han people and 7 directly-entering-socialism groups, which were Va, Dai, Yi, Ladhull, Bai, Blang, and Yao in DESEC. And most of middle school students were Va people (84.4%). Most of students' guardians were farmer (88.7%) coming from rural region.

### Demographic characteristics of respondents' experiences of blended learning

Table 3 revealed that whiteboard (67.7%), phone (53.4%) and computer (40%) were the three most accessible devices for blended learning (n = 635). Junior high school students' blended learning experience period were more than 1 years. Most of students (55%) experienced blended learning for less than 3 years. There was still part of students reported that they hardly had time studying online in the classroom (22.9%) or after school (19%). Most students only could spend 0.5–1 hour studying online both in classroom (50.6%) and after school (45%).

### Descriptive statistics of measurement scales

According to Cho & Teo, five level for evaluating the mean value of students' level of agreement on each of item in the 5-point Likert scale were interpreted Strongly Agree (4.21–5.00),

**Table 2. Demographic characteristics of respondents.**

| Characteristic | Variables | Frequency | n (%) |
|---|---|---|---|
| Grade | K8 | 341 | 53.7 |
| | K9 | 294 | 46.3 |
| Gender | Male | 319 | 50.2 |
| | Female | 316 | 49.8 |
| Ethnicity | Va | 365 | 84.4 |
| | Han | 56 | 8.8 |
| | Dai | 33 | 5.2 |
| | Yi | 5 | 0.8 |
| | Ladhull | 2 | 0.3 |
| | Bai | 1 | 0.2 |
| | Blang | 1 | 0.2 |
| | Yao | 1 | 0.2 |
| Occupation of Guardians | Farmer | 563 | 88.7 |
| | Self-employed | 45 | 7.1 |
| | National staff | 16 | 2.5 |
| | Others | 11 | 1.7 |

**Table 3. Respondents' experiences on blended learning.**

| Characteristic | Variables | Frequency | n (%) |
|---|---|---|---|
| Instruments | whiteboard | 430 | 67.7 |
| | phone | 339 | 53.4 |
| | computer | 254 | 40 |
| | pad | 90 | 14.2 |
| | others | 4 | 0.6 |
| Blended learning experience period | n<1 year | 0 | 0 |
| | 1≤n≤2 years | 266 | 41.9 |
| | 2<n≤3 years | 83 | 13.1 |
| | More than 3 years | 286 | 45 |
| Study hours in an online classroom | Almost no time | 146 | 22.9 |
| | 0.5≤n≤1 hour | 321 | 50.6 |
| | 2≤n<3 hours | 107 | 16.9 |
| | n≥3 hours | 61 | 9.6 |
| Study hours via online after school | Almost no time | 123 | 19.4 |
| | 0.5≤n≤1 hour | 286 | 45 |
| | 2≤n<3 hours | 178 | 28 |
| | n≥3 hours | 48 | 7.6 |

Agree (3.41–4.20), Uncertain (2.61–3.40), Disagree (1.81–2.60), and Strongly Disagree (0.00–1.80) [45]. The interpretation of students' level of agreement on 6 variables were all "Agree"(3.76–3.89), as shown in Table 4.

## Reliability and validity

In the study, six dimensions' Cronbach's Alpha ranged from 0.863 to 0.889 (see Table 5). It indicated that the intrinsic consistency reliability of the scale questions was satisfactory [46]. CITC values of items in this study shown as followed table ranged from 0.648 to 0.797, which was higher than 0.50 [47]. Thus, the instrument was reliable. All variable correlations in the correlation matrix were more than 0.3, the measure of sampling adequacy (MAS) values of a single variable on the diagonal of the reflection image correlation matrix in the study were greater than 0.5. Kaiser-Meyer-Olkin (KMO) value was 0.960, which meant very suitable when the value was higher than 0.9 [48], and the approximate chi-square of the Bartlett's Test of Sphericity was large, and the p-value was significant ($X^2$ = 9342.4, p< 0.01) indicated that correlations between items were sufficiently large [49]. All index met the conditions for factor analysis.

The result of CFA found that the factor loading values greater than 0.5 [50]. AVE value exceeded 0.50 which meant it was adequate for convergent validity. The CR values exceeded the limit of 0.6 for achieving convergent validity. The value of Critical Ratio (C.R.) is equal to the ratio between Estimate and the Standard Error of Estimate, which is equivalent to the T-value. If C.R. is higher than 1.96, the parameter estimate value reaches a significant level of 0.05 [51]. All significances were at the 0.001 significant levels (p<0.001). Therefore, it proved that all constructs were reliable and validated for further analysis.

Table 6 showed that the square root of the average variance extracted value was larger than the corresponding correlation coefficient, indicating that the discriminant validity of scale in this study was good [52].

**Table 4. Descriptive statistics of measurement scales.**

| Dimensions Item | | Mean | Standard Deviation | Interpretation |
|---|---|---|---|---|
| **Perceived Usefulness** | PU1 | 3.92 | .698 | Agree |
| | PU2 | 3.89 | .695 | Agree |
| | PU3 | 3.81 | .744 | Agree |
| | PU4 | 3.91 | .694 | Agree |
| **Average** | | **3.88 Agree** | | |
| **Satisfaction** | S1 | 3.88 | .691 | Agree |
| | S2 | 3.88 | .705 | Agree |
| | S3 | 3.90 | .673 | Agree |
| | S4 | 3.90 | .667 | Agree |
| **Average** | | **3.89 Agree** | | |
| **Confirmation** | C1 | 3.78 | .751 | Agree |
| | C2 | 3.78 | .724 | Agree |
| | C3 | 3.79 | .757 | Agree |
| | C4 | 3.69 | .777 | Agree |
| **Average** | | **3.76 Agree** | | |
| **Self-efficacy** | SE1 | 3.77 | .744 | Agree |
| | SE2 | 3.80 | .729 | Agree |
| | SE3 | 3.75 | .749 | Agree |
| | SE4 | 3.72 | .760 | Agree |
| **Average** | | **3.76 Agree** | | |
| **Information quality** | IQ1 | 3.85 | .699 | Agree |
| | IQ2 | 3.79 | .699 | Agree |
| | IQ3 | 3.85 | .657 | Agree |
| | IQ4 | 3.80 | .687 | Agree |
| | IQ5 | 3.75 | .715 | Agree |
| **Average** | | **3.81 Agree** | | |
| **Continuance Intention** | CI1 | 3.99 | .745 | Agree |
| | CI2 | 3.81 | .805 | Agree |
| | CI3 | 3.90 | .717 | Agree |
| | CI4 | 3.82 | .738 | Agree |
| **Average** | | **3.88 Agree** | | |

## Model fit and model result

Structural equation modeling (SEM) result was presented in Table 7. All the parameters, i.e. CMIN/df (= 2.974<3.0), GFI (= 0.918>0.90), AGFI (= 0.892>0.85), CFI (= 0.958 >0.95), NFI (= 0.938>0.90) and RMSEA (= 0.056 <0.08) met the standard. It indicated that the model was acceptable.

## Hypothesis testing

The result of the structure model including standardized path coefficients($\beta$), t-values, p-value, and explained variances ($R^2$) was shown in Fig 2. The higher the value of $R^2$, the higher the degree of interpretation, and the better the model. When $R^2$, at least, is equal to or higher than 0.1($R^2 \geq 0.1$), the model explaining Continuance Intention ($R^2$ = 0.665), Satisfaction ($R^2$ = 0.766) and Perceived usefulness($R^2$ = 0.718) were acceptable [53]. It indicates the research model in this study gained good explanatory power. Perceived usefulness ($\beta$ = 0.593, p < 0.001) significantly impacted continuance intention [54–56]. Satisfaction ($\beta$ = 0.397,

**Table 5. Reliability and validity of measurement model.**

| Dimension | CITC >0.5 | Factor Loading >0.5 | Cronbach's alpha | C.R. >1.96 & P-value<0.5 | CR >0.6 | AVE >0.5 |
|---|---|---|---|---|---|---|
| Perceived Usefulness | | | .880 | | 0.879 | 0.645 |
| PU1 | .736 | .777 | | | | |
| PU2 | .724 | .773 | | 19.240*** | | |
| PU3 | .731 | .802 | . | 19.666*** | | |
| PU4 | .771 | .857 | . | 21.094*** | | |
| Satisfaction | | | .889 | | 0.890 | 0.667 |
| S1 | .739 | .809 | | | | |
| S2 | .736 | .782 | | 22.056*** | | |
| S3 | .763 | .842 | | 24.424*** | | |
| S4 | .784 | .837 | | 24.225*** | | |
| Confirmation | | | .887 | | 0.889 | 0.666 |
| C1 | .758 | .824 | | | | |
| C2 | .797 | .866 | | 26.193*** | | |
| C3 | .762 | .811 | | 23.806*** | | |
| C4 | .695 | .761 | | 21.781*** | | |
| Self-efficacy | | | .874 | | 0.871 | 0.629 |
| SE1 | .744 | .835 | | | | |
| SE2 | .759 | .857 | | 25.742*** | | |
| SE3 | .722 | .753 | | 21.363*** | | |
| SE4 | .695 | .718 | | 19.977*** | | |
| Information Quality | | | .863 | | 0.863 | 0.559 |
| IQ1 | .706 | .791 | | | | |
| IQ2 | .648 | .719 | | 19.240*** | | |
| IQ3 | .685 | .729 | | 19.666*** | | |
| IQ4 | .717 | .774 | | 21.094*** | | |
| IQ5 | .656 | .722 | | 19.337*** | | |
| Continuance Intention | | | .873 | | 0.881 | 0.648 |
| CI1 | .700 | .789 | | | | |
| CI2 | .735 | .783 | | 20.789*** | | |
| CI3 | .773 | .820 | | 21.939*** | | |
| CI4 | .710 | .828 | | 20.134*** | | |

CITC = Corrected Item-Total Correlation, CR = Composite Reliability, AVE = Average Variance Extracted, C.R. (T-value) = Critical Ratio

*** = Significant at the 0.001 significant levels (p<0.001).

**Table 6. Discriminative validity analysis result.**

| √AVE | PU | S | C | SE | IQ | CI |
|---|---|---|---|---|---|---|
| PU | 0.803 | | | | | |
| S | 0.720 | 0.817 | | | | |
| C | 0.641 | 0.716 | 0.816 | | | |
| SE | 0.669 | 0.705 | 0.753 | 0.793 | | |
| IQ | 0.672 | 0.726 | 0.729 | 0.729 | 0.748 | |
| CI | 0.655 | 0.651 | 0.709 | 0.701 | 0.740 | 0.805 |

**Table 7. Fit Indices for the structural model.**

| Statistical Values Obtained from Analysis | | | |
|---|---|---|---|
| **Index** | **Criterion** | | **value** |
| | acceptable | good | |
| **CMIN/df** | $< 3.0$ | $< 2.0$ | 2.974 |
| **GFI** | $\geq 0.90$ | $\geq 0.95$ | 0.918 |
| **AGFI** | $\geq 0.85$ | $\geq 0.90$ | 0.892 |
| **CFI** | $\geq 0.95$ | $\geq 0.97$ | 0.958 |
| **NFI** | $\geq 0.90$ | $\geq 0.95$ | 0.938 |
| **RMSEA** | $< 0.08$ | $< 0.05$ | 0.056 |
| **Model Summary** | | | Harmony |

p < 0.001) significantly impacted continuance intention [14,57,58]. Perceived usefulness (β = 0.317, p < 0.001) significantly affected satisfaction [59,60]. Confirmation (β = 0.567, p < 0.01) significantly affected satisfaction [14,61]. Information quality (β = 0.632, p < 0.001) significantly affected perceived usefulness [61,62]. Self-efficacy (β = 0.384, p < 0.001) significantly affected perceived usefulness [63]. However, confirmation(T-value = -1.401, p>0.05) did not statistically significantly affect perceived usefulness [62,64]. This is contrary to previous findings [25,65]. Hence, $H_{01}$, $H_{02}$, $H_{03}$, $H_{05}$, $H_{06}$, and $H_{07}$ were supported. $H_{04}$ were not supported, thus the study accepted $H_{a4}$.

Total effect (TE) is the accumulation of direct and indirect effects of a relationship path [66]. Direct, indirect, and total effects of relationships between variables were shown in Table 8. Total effects imposed on Satisfaction of blended learning, from strongest to weakest, was Confirmation (0.511***), Perceived Usefulness (0.337***), information Quality (0.211***), Self-efficacy (0.118***). Total effects exerted on Continuance Intention of blended learning, from most to least, was Perceived Usefulness (0.645***), Satisfaction (0.406***), Information Quality (0.404***), Self-efficacy (0.226***) and Confirmation (0.145***). Total effect of information Quality (0.628***) imposed on Perceived Usefulness was higher than Self-efficacy (0.350***).

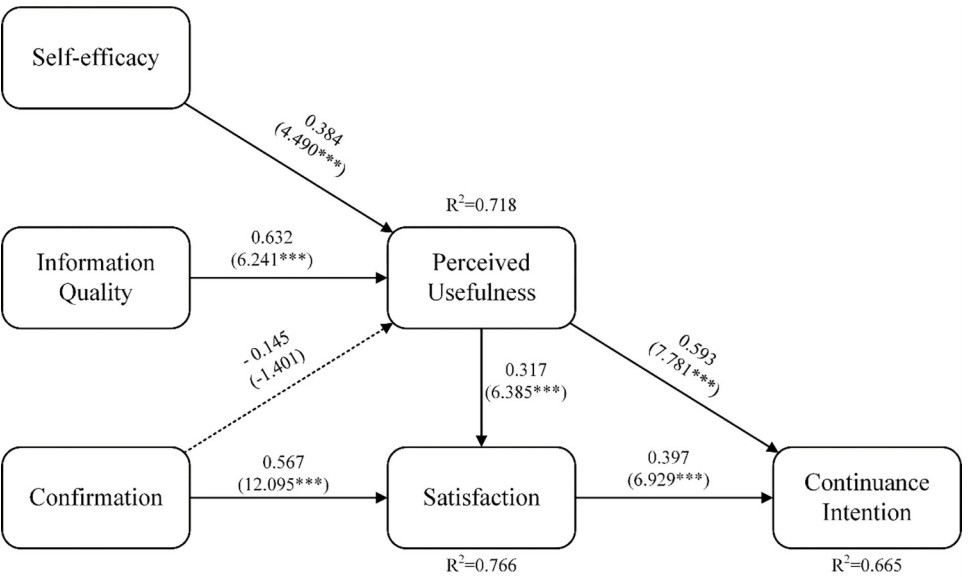

**Fig 2. Result of the structural model.** T-values and p-value are in parentheses, ***p<0.001.

**Table 8. Direct, indirect, and total effects of relationships.**

| Effect | | | Independent variables (IV) | | | | |
|---|---|---|---|---|---|---|---|
| | | | SE | IQ | C | PU | S |
| **Dependent variables (DV)** | **PU** | DE | 0.350 *** | 0.629 *** | -0.123 | - | - |
| | | IE | - | - | - | - | - |
| | | TE | 0.350 *** | 0.626 *** | -0.123 | - | - |
| | | R² | | | 0.718 | | |
| | **S** | DE | - | - | 0.553 *** | 0.337 *** | - |
| | | IE | 0.118 *** | 0.211 *** | -0.041 | - | - |
| | | TE | 0.118 *** | 0.211 *** | 0.511 *** | 0.337 *** | - |
| | | R² | | | 0.766 | | |
| | **CI** | DE | - | - | - | 0.508 *** | 0.406 *** |
| | | IE | 0.226 *** | 0.404 *** | 0.145 *** | 0.137 *** | - |
| | | TE | 0.226 *** | 0.404 *** | 0.145 *** | 0.645 *** | 0.406 *** |
| | | R² | | | 0.665 | | |

IV = Independent Variables, DV = Dependent Variables, CI = Continuance Intention, S = Satisfaction, PU = Perceived Usefulness, C = Confirmation, IQ = Information Quality, SE = Self-efficacy, DE = Direct Effect, IE = Indirect Effect, TE = Total Effect (DE+IE)

***p<0.001

**p<0.01 and

*p<0.05.

## Discussion

Instrument for blended learning was insufficient, especially mobile device in classroom in DESEC. Most students spent little time studying online. Why students face-to-face learning accounts for the majority of blended learning. And students' agreement on each of item were interpreted as "Agree", but not "Strongly Agree". All conditions above limited junior students' perception on blended learning in DESEC.

In this study, perceived usefulness was the most important factor affecting continuance intention, which is not consistent with most of previous studies proposing that satisfaction was the strongest predictor of continuance intention [27]. It indicated that usefulness overwhelms satisfaction as long as it improves students' academic performance in DESEC where students' performance was only evaluated by test score in exam-oriented assessment system.

Information quality was the most important antecedent acting on junior high students' perceived usefulness of blended learning in DESEC, which further explained junior high students' satisfaction and continuance intention of blended learning. Providing higher information quality increases the perceived usefulness by enhancing the fit between content and user's information requirements [67]. Nowadays, the content of resources online is more focused on "teaching", and there is a serious shortage of high-quality information for students to learn independently in rural areas. What' more, these resources only focus on students' academic performance, and almost ignore moral education, physical education, art and labor education in China [7]. Thus, the critical direction of enhancing blended learning is to provide junior

high students in DESEC with diverse learning resources based on their characteristics, interests and needs, and make prestigious teachers' classes easily accessible through online technology.

Confirmation was the most important antecedent acting on junior high students' satisfaction of blended learning in DESEC in this model. However, confirmation did not affect perceived usefulness. Bhattacherjee gave the hypothesis based on the cognitive dissonance theory. It suggests if user pre acceptance usefulness perceptions are disconfirmed during actual use, they may experience cognitive disorder [18]. Then users may remedy this dissonance by elevating their perceived usefulness. Further studies need to be conducted to explain the association between confirmation and perceived usefulness.

With technologies advancing rapidly, machines will replace humans in experience-based predictions, decision making, optimizations, data analysis, and others [68]. Machine learning is becoming essential for teaching and learning. In this way, by providing the learning algorithm with experience data, the model in this study will be used to train machine predicting students' satisfaction and enhancing their continuance intention of blended learning in the future [69–71].

## Conclusion

The study enhances the understanding of blended learning of middle school students on their satisfaction and intention to continue blended learning in DESEC. Perceived usefulness is the strongest predictor of continuance intention, while Confirmation was the strongest predictor acting on junior high students' satisfaction of blended learning in DESEC. Perception of blended learning for junior high students formed, but is not strong in DESEC. Instrument for blended learning is insufficient, especially mobile device in classroom. Perceived usefulness overwhelms satisfaction as the strongest predictor of continuance intention, and traditional learning (face to face) still takes advantage for an exam-oriented assessment system in DESEC.

## Acknowledgments

The authors would like to acknowledge the efforts of the data collectors involved in the study.

## Author Contributions

**Conceptualization:** Chunyu Li, Thanawan Phongsatha.

**Data curation:** Chunyu Li.

**Formal analysis:** Chunyu Li.

**Funding acquisition:** Chunyu Li.

**Investigation:** Chunyu Li.

**Methodology:** Chunyu Li.

**Project administration:** Chunyu Li, Thanawan Phongsatha.

**Resources:** Chunyu Li.

**Software:** Chunyu Li.

**Supervision:** Chunyu Li, Thanawan Phongsatha.

**Validation:** Chunyu Li, Thanawan Phongsatha.

**Visualization:** Chunyu Li.

**Writing – original draft:** Chunyu Li.

**Writing – review & editing:** Chunyu Li.

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
