## [Decision Letter · Decision Letter 0]

29 Aug 2022

PONE-D-22-16934Satisfaction and continuance intention of blended learning from perspective of junior high school students in the directly-entering-socialism ethnic communities of ChinaPLOS ONE

Dear Authors

Thank you for submitting your manuscript to PLOS ONE. After careful consideration, we feel that it has merit but does not fully meet PLOS ONE’s publication criteria as it currently stands. Therefore, we invite you to submit a revised version of the manuscript that addresses the points raised during the review process.

ACADEMIC EDITOR:Comments to Author:The authors are required to submit the revision. 

We look forward to receiving your revised manuscript.

Kind regards,

Dr. Mudassir Khan, Ph.D

Academic Editor

PLOS ONE

Journal Requirements:

2. PLOS ONE does not copy edit accepted manuscripts (https://journals.plos.org/plosone/s/criteria-for-publication#loc-5). To that effect, please ensure that your submission is free of typos and grammatical errors.

5.  We note you have included a table to which you do not refer in the text of your manuscript. Please ensure that you refer to Table 7 in your text; if accepted, production will need this reference to link the reader to the Table.

Additional Editor Comments:

The authors have proposed a blended learning process from perspective of junior

high school students in the directly-entering-socialism ethnic communities of China. The reviewer appreciate the authors efforts. However, following comments need to be addressed:

1. In Section Previous research and research model, what is "[Error! Reference source not found.]"

2. The label of "Figure 2. Result of the Structural Model" is not properly placed.

3. In table 1, more recent papers need to be added.

Reviewers' comments:

Reviewer's Responses to Questions

**Comments to the Author**

1. Is the manuscript technically sound, and do the data support the conclusions?

Reviewer #1: Partly

Reviewer #2: Yes

Reviewer #3: Yes

2. Has the statistical analysis been performed appropriately and rigorously? 

Reviewer #1: Yes

Reviewer #2: Yes

Reviewer #3: Yes

3. Have the authors made all data underlying the findings in their manuscript fully available?

Reviewer #1: Yes

Reviewer #2: Yes

Reviewer #3: Yes

4. Is the manuscript presented in an intelligible fashion and written in standard English?

Reviewer #1: No

Reviewer #2: Yes

Reviewer #3: Yes

5. Review Comments to the Author

Reviewer #1: Introduction section is very short, Motivation of the work should be added in the Introduction and also challenges in the current system.

proposed methodology could be represent in the scientific manner.

Its good authors have provided the sufficient results however the representation of the manuscript need to be done.

Latest work should be added, Update the conclusion & abstract accordingly.

Rashid, E., Ansari, M. D., Gunjan, V. K., & Khan, M. (2020). Enhancement in teaching quality methodology by predicting attendance using machine learning technique. In Modern approaches in machine learning and cognitive science: a walkthrough (pp. 227-235). Springer, Cham.

Sethi, K., Jaiswal, V., & Ansari, M. D. (2020). Machine learning based support system for students to select stream (subject). Recent advances in computer science and communications (Formerly: Recent patents on computer science), 13(3), 336-344.

Rashid, E., Ansari, M., & Gunjan, V. K. (2022). Innovation and Entrepreneurship in the Technical Education. In ICCCE 2021 (pp. 1199-1203). Springer, Singapore.

Reviewer #2: The authors have proposed a blended learning process from perspective of junior

high school students in the directly-entering-socialism ethnic communities of China. The reviewer appreicate the authors efforts. However, following comments need to be addressed:

1. In Section Previous research and research model, what is "[Error! Reference source not found.]"

2. The label of "Figure 2. Result of the Structural Model" is not properly placed.

3. In table 1, more recent papers need to be added.

Reviewer #3: Paper is based on survey.

A satisfactory amount of data has been collected and analyzed.

Results are well organized and calculated.

Conclusion parts meets the objective of the research statement.

References are sufficient and latest

6. PLOS authors have the option to publish the peer review history of their article (what does this mean?). If published, this will include your full peer review and any attached files.

Reviewer #1: No

Reviewer #2: **Yes: **Dr. Mohd. Naseem

Reviewer #3: No

---

## [Author Response · Author response to Decision Letter 0]

10 Oct 2022

Dear Editor, and Dear Reviewers,

We would like to thank you for your comments and suggestions, which gave us a lot of help to improve the quality of manuscript. In the revised manuscript, we revised all the issues raised. In this document, we answered all the questions proposed by journal, editor and the reviewers. Comments are shown in black bold font, followed by our answer/comment in blue normal font. The major corrections/changes in the manuscript are displayed in red font and blue front.

Journal Requirements:

Answer:

Yes, we have made manuscript meet PLOS ONE's style requirements, including those for file naming. What’s more, pages and line numbers were added. Abstract was modified to 288 words(not exceed 300 words).

2. PLOS ONE does not copy edit accepted manuscripts. To that effect, please ensure that your submission is free of typos and grammatical errors.

Answer:

Yes, we have checked the words and grammar ensuring the manuscript is free of typos and grammatical errors. 

Answer:

We wish to make changes to our Data Availability statement, we will describe these changes in our cover letter. The data that support the findings of this study are available on request from the corresponding author.

Answer:

We have amended abstract both identical.

5. We note you have included a table to which you do not refer in the text of your manuscript. Please ensure that you refer to Table 7 in your text; if accepted, production will need this reference to link the reader to the Table.

Answer:

We have amended the mistakes in the text. You can check it on line numbers 312 in revised manuscript with track changes.

Answer:

We have amended reference format as outlined in the ICMJE sample references as PLOS ONE requires. And we added the updated references in order as reviewers comments. Details was described as follow:

For introduction part, we added four references from 6 to 9 in reference list (line number 95-99 in the text). 

For table1, we added fourteen references from 30 to 43 in reference list (line number 202-208 in the text).

For discussion part, we added four references from 68 to 71 in reference list (line number 383-386 in the text).

Additional Editor Comments:

The authors have proposed a blended learning process from perspective of junior high school students in the directly-entering-socialism ethnic communities of China. The reviewer appreciate the authors efforts. However, following comments need to be addressed:

1. In Section Previous research and research model, what is "[Error! Reference source not found.]"

Answer:

We have amended this mistake on line number 149. It was a wrong hyperlink which displayed error in PDF file.

2. The label of "Figure 2. Result of the Structural Model" is not properly placed.

Answer:

We have amended this mistake on line number 333. 

3. In table 1, more recent papers need to be added.

Answer:

We have added more recent papers.

Perceived Usefulness: Almaiah et al. (2019), Alkhawaja et al. (2022).

Satisfaction: Almaiah et al. (2019), Warren et al. (2020)

Confirmation: Yang et al. (2022)

Self-efficacy: Vermunt& Donche (2017), Yang et al. (2022)

Information Quality: Almaiah et al. (2019), Alfaki et al. (2021)

Continuance Intention: Li et al. (2021), Yang et al. (2022)

Review Comments:

Reviewer #1: 

Introduction section is very short, Motivation of the work should be added in the Introduction and also challenges in the current system.

proposed methodology could be represent in the scientific manner.

Its good authors have provided the sufficient results however the representation of the manuscript need to be done.

Latest work should be added, Update the conclusion & abstract accordingly.

Rashid, E., Ansari, M. D., Gunjan, V. K., & Khan, M. (2020). Enhancement in teaching quality methodology by predicting attendance using machine learning technique. In Modern approaches in machine learning and cognitive science: a walkthrough (pp. 227-235). Springer, Cham.

Sethi, K., Jaiswal, V., & Ansari, M. D. (2020). Machine learning based support system for students to select stream (subject). Recent advances in computer science and communications (Formerly: Recent patents on computer science), 13(3), 336-344.

Rashid, E., Ansari, M., & Gunjan, V. K. (2022). Innovation and Entrepreneurship in the Technical Education. In ICCCE 2021 (pp. 1199-1203). Springer, Singapore.

Answer:

We thank you for the general appreciation of our work, and specific comments and references given that help to improve our manuscript. 

We added the motivation of the work and challenges in the current system in the Introduction as mentioned. For the result presentation, we divided it into four parts, i.e. demographic information, reliability and validity, model fit, and hypothesis testing. For each part, we described the data of the result following a figure. And Proposed methodology was modified. 

We thank you for giving us the three references to review. It inspires us a lot for the further study. We read more articles and books on machine learning. We updated and added these references in the Discussion considering that the conclusion and abstract should keep objectively described focused on the research objectives after intense discussion and deep thinking. In the discussion, we think the research model will hopefully be used in machine learning to predict student satisfaction and continuance intention of blended learning with experience data.

Reviewer #2: 

The authors have proposed a blended learning process from perspective of junior high school students in the directly-entering-socialism ethnic communities of China. The reviewer appreicate the authors efforts. However, following comments need to be addressed:

1. In Section Previous research and research model, what is "[Error! Reference source not found.]"

2. The label of "Figure 2. Result of the Structural Model" is not properly placed.

3. In table 1, more recent papers need to be added.

Answer:

We thank you for recognition of our efforts, and specific comments. 

We Revised accordingly. 

For "[Error! Reference source not found.]",We have amended this mistake on line number 149. It was a wrong hyperlink which displayed error in PDF file.

For the label of "Figure 2", we replaced as the journal requires.

For table 1, we have added more recent papers.

Perceived Usefulness: Almaiah et al. (2019), Alkhawaja et al. (2022).

Satisfaction: Almaiah et al. (2019), Warren et al. (2020)

Confirmation: Yang et al. (2022)

Self-efficacy: Vermunt& Donche (2017), Yang et al. (2022)

Information Quality: Almaiah et al. (2019), Alfaki et al. (2021)

Continuance Intention: Li et al. (2021), Yang et al. (2022)

Reviewer #3:

Paper is based on survey.

A satisfactory amount of data has been collected and analyzed.

Results are well organized and calculated.

Conclusion parts meets the objective of the research statement.

References are sufficient and latest.

Answer:

We thank you for the specific comments of our manuscript. 

Beast regards

Chunyu li

---

## [Decision Letter · Decision Letter 1]

15 Nov 2022

Satisfaction and continuance intention of blended learning from perspective of junior high school students in the directly-entering-socialism ethnic communities of China

PONE-D-22-16934R1

Dear Chunyu Li,

We’re pleased to inform you that your manuscript has been judged scientifically suitable for publication and will be formally accepted for publication once it meets all outstanding technical requirements.

Kind regards,

Mudassir Khan, Ph.D

Academic Editor

PLOS ONE

Additional Editor Comments (optional):

Accepted

Reviewers' comments:

Reviewer's Responses to Questions

**Comments to the Author**

1. If the authors have adequately addressed your comments raised in a previous round of review and you feel that this manuscript is now acceptable for publication, you may indicate that here to bypass the “Comments to the Author” section, enter your conflict of interest statement in the “Confidential to Editor” section, and submit your "Accept" recommendation.

Reviewer #1: All comments have been addressed

Reviewer #2: All comments have been addressed

2. Is the manuscript technically sound, and do the data support the conclusions?

Reviewer #1: Yes

Reviewer #2: Yes

3. Has the statistical analysis been performed appropriately and rigorously? 

Reviewer #1: Yes

Reviewer #2: Yes

4. Have the authors made all data underlying the findings in their manuscript fully available?

Reviewer #1: Yes

Reviewer #2: Yes

5. Is the manuscript presented in an intelligible fashion and written in standard English?

Reviewer #1: Yes

Reviewer #2: Yes

6. Review Comments to the Author

Reviewer #1: Authors have incorporated the comments and suggestion, I have no further comments. It may be accepted for publications.

Reviewer #2: My concerns have been addressed in the current form of the manuscript. I would recommend to accept this manuscript.

7. PLOS authors have the option to publish the peer review history of their article (what does this mean?). If published, this will include your full peer review and any attached files.

Reviewer #1: No

Reviewer #2: **Yes: **Mohd. Naseem

---

## [Editor Report · Acceptance letter]

21 Nov 2022

PONE-D-22-16934R1 

Satisfaction and continuance intention of blended learning from perspective of junior high school students in the directly-entering-socialism ethnic communities of China 

Dear Dr. Li:

I'm pleased to inform you that your manuscript has been deemed suitable for publication in PLOS ONE. Congratulations! Your manuscript is now with our production department. 

Kind regards, 

on behalf of

Dr. Mudassir Khan 

Academic Editor

PLOS ONE